# Cavitation upon low-speed solid–liquid impact

Nathan B. Speirs [1✉], Kenneth R. Langley [1✉], Zhao Pan[2✉], Tadd T. Truscott [1✉] &
Sigurdur T. Thoroddsen[1✉]

When a solid object impacts on the surface of a liquid, extremely high pressure develops at
the site of contact. Von Karman's study of this classical physics problem showed that the
pressure on the bottom surface of the impacting body approaches infinity for flat impacts.
Yet, in contrast to the high pressures found from experience and in previous studies, we show
that a flat-bottomed cylinder impacting a pool of liquid can decrease the local pressure
sufficiently to cavitate the liquid. Cavitation occurs because the liquid is slightly compressible
and impact creates large pressure waves that reflect from the free surface to form negative
pressure regions. We find that an impact velocity as low as ~3 m/s suffices to cavitate the
liquid and propose a new cavitation number to predict cavitation onset in low-speed solid-
liquid impact-scenarios. These findings imply that localized cavitation could occur in impacts
such as boat slamming, cliff jumping, and ocean landing of spacecraft.

[1] Division of Physical Sciences and Engineering, King Abdullah University of Science and Technology (KAUST), Thuwal 23955-6900, Saudi Arabia.
[2] Department of Mechanical and Mechatronics Engineering, University of Waterloo, Waterloo N2L 3G1 ON, Canada. ✉email: nathanspeirs@gmail.com;
kenrlangley@gmail.com; zhao.pan@uwaterloo.ca; taddtruscott@gmail.com; sigurdur.thoroddsen@kaust.edu.sa

The study of a blunt object slamming into a large body of water dates back to the 1920s when von Karman[1] analytically studied the impact pressure on the bottom of wedge-shaped seaplane floats. Soon thereafter, Wagner[2] expanded on von Karman's theory, and continuing efforts since have developed more complex incompressible[3] and compressible[4,5] models. One of these models hypothesizes that the interaction of the pressure waves emitted when a blunt body impacts on a free surface could create negative pressure zones that cavitate the liquid[5]. Some relevant work has shown other scenarios in which free surface impact causes cavitation. For example, a liquid jet impacting a liquid pool at 600 m/s can create a cloud of cavitation bubbles beneath it[6,7]. A bullet shot horizontally at 200–500 m/s at a falling liquid jet[8,9] and droplets impacting solid surfaces at 20–110 m/s[10–12] can also cause cavitation after reflection and focusing of the pressure waves from the opposing gas-liquid interface. These relevant experiments confirm that cavitation due to free surface impact can occur at very high impact velocity, where the compressibility of water is appreciable.

Our experiments show that cavitation can occur at much lower velocities (on the order of 1 m/s), when a flat-bottomed cylinder impacts onto a liquid pool, as shown in Fig. 1. The sudden impact compresses the liquid beneath, and the resulting pressure waves emanate outward at the speed of sound in the liquid ($c = 1480$ m/s in water) forming a cloud of cavitation bubbles in the wake of the negative pressure wave. This type of cavitation is caused by liquid compression and pressure wave reflection and hence is fundamentally different from acceleration-induced cavitation which is also seen at low flow velocities[13–15].

## Results

To form a more complete understanding of this impact scenario, we examine time-resolved images from synchronized high-speed cameras and pressure measurements from hydrophones. Figure 2a and b show two views of a cylinder with diameter

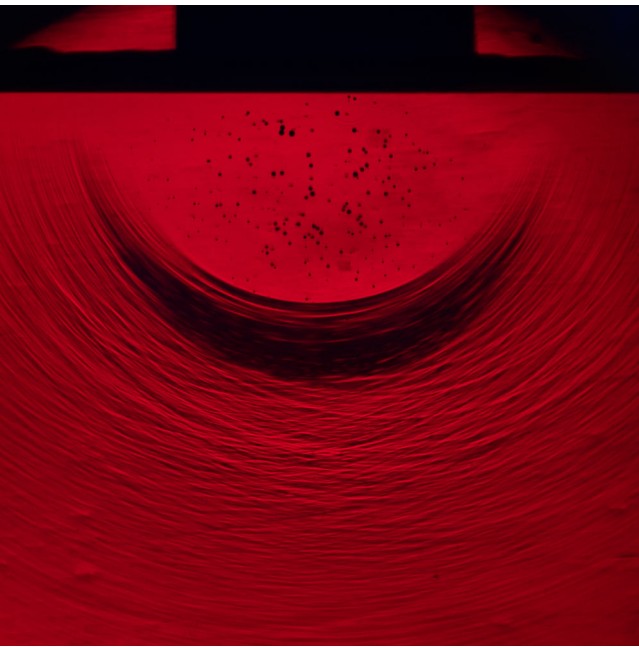

**Fig. 1 This schlieren image shows the impact of a 20-mm-diameter, flat-bottom cylinder on a pool of water as it falls downward with a velocity of ~ 9 m/s.** The impact creates intense compression and tension waves, the gradients of which are shown by the bright and dark striations, that cause the water to cavitate within ~100 μs of initial contact. The red coloring is from the laser lighting.

$d = 30$ mm impacting at $U_o = 9.38$ m/s at a slight angle (between the water surface and the cylinder bottom) of $\alpha = 0.37°$. Before the cylinder contacts the pool, it compresses the air between itself and the pool forming a depression in the free surface, as seen in Fig. 2a at $t = -7.1$ μs. The cylinder initially touches the water on the left (at $t = 0$) forming a contact line that moves to the right (Fig. 2a $t = 3.2$–13.5 μs) at a velocity of $U_{cl} = U_o/\alpha \approx 1455$ m/s (using the small angle approximation). As the contact line moves, a compression wave forms with it, increasing the pressure in the liquid beneath the cylinder as indicated by the bright region in Fig. 2b at $t = 14$ μs. When the compression wave reaches the right edge of the cylinder, it reflects off the pool's surface forming a tension wave (dark region in the upper right) seen in Fig. 2b at $t = 22$ μs. Similar reflections occur along the entire circumference of the cylinder as the contact line moves across the bottom of the cylinder. This forms multiple small tension waves that propagate out radially from the point of reflection creating a Doppler-shifted wave pattern shown by the dark striations in Fig. 2b at $t = 14-22$ μs. The tension waves overlap and sum together on the right half of the cylinder forming a negative pressure zone shown by the large dark band that propagates down into the pool at $t = 30-46$ μs. This large tension wave initiates cavitation just beneath the right edge of the cylinder where last contact occurs (Fig. 2a at $t = 23.8-34.2$ μs) and forms a large conical cloud of cavitation bubbles as the waves propagate down into the pool as shown in Fig. 2b at $t = 30-54$ μs. As the cavitation bubbles collapse they form spherical shock waves that grow radially outward ($t = 38-54$ μs). A hydrophone signal of this event (Fig. 2d, red line), shows the same pattern in the pressure, revealing a maximum of 2.49 atm followed by a rapid drop to $-2.66$ atm as measured 96 mm away from the impact location where the large pressure amplitudes of the near impact region have diminished.

Just a slight increase in the impact angle $\alpha$ drastically changes the pressure field and inhibits cavitation. The impact shown in Fig. 2c is very similar to that shown in a and b except that $\alpha$ has increased to 1.09° and the orientation relative to the camera has shifted such that the front right of the cylinder contacts the pool first and the contact line moves away from the viewer towards the back left. The larger $\alpha$ generates less intense pressure waves (Fig. 2c, $t = 32$ μs), a slower moving contact line ($U_{cl} = 480$ m/s), and less summing of the tension regions (Fig. 2c, $t = 60$ μs). This results in low-amplitude-pressure oscillations (Fig. 2d, blue line) and no cavitation (Fig. 2c, $t = 74$ μs).

Cavitation onset is typically predicted by a canonical non-dimensional group called the cavitation number[16–18]:

$$Ca_v = \frac{P_r - P_c}{P_d} = \frac{P_{amb} - P_v}{\frac{1}{2}\rho v^2},\qquad(1)$$

where $P_r$ is the reference or background pressure often set equal to the ambient pressure $P_{amb}$; $P_c$ is the pressure at which cavitation occurs and is typically considered to be the liquid vapor pressure $P_v$; $P_d = \frac{1}{2}\rho v^2$ is the dynamic pressure drop due to high flow speed $v$ of the liquid with density $\rho$. When the pressure drop is higher than the difference between $P_{amb}$ and $P_v$ (i.e., $Ca < 1$) cavitation is expected to happen[16–18]. Yet, according to this canonical formulation, our experiments show the onset of cavitation to occur over a large range of $Ca_v$; from $O(10^{-3})$ to $O(10^1)$ when using the impact velocity $U_o$ as the characteristic velocity and from $O(10^{-5})$ to $O(10^{-4})$ when using the contact line velocity $U_{cl}$. This shows that neither velocity consistently predicts transition from cavitating to non-cavitating impacts. Eq. (1) also indicates that the ambient pressure should affect the onset of cavitation. Yet, we find that at the high end of our velocity range, the onset of cavitation is independent of the ambient pressure, which we control here in a vacuum chamber. At lower impact velocities and small angles we find that the ambient pressure

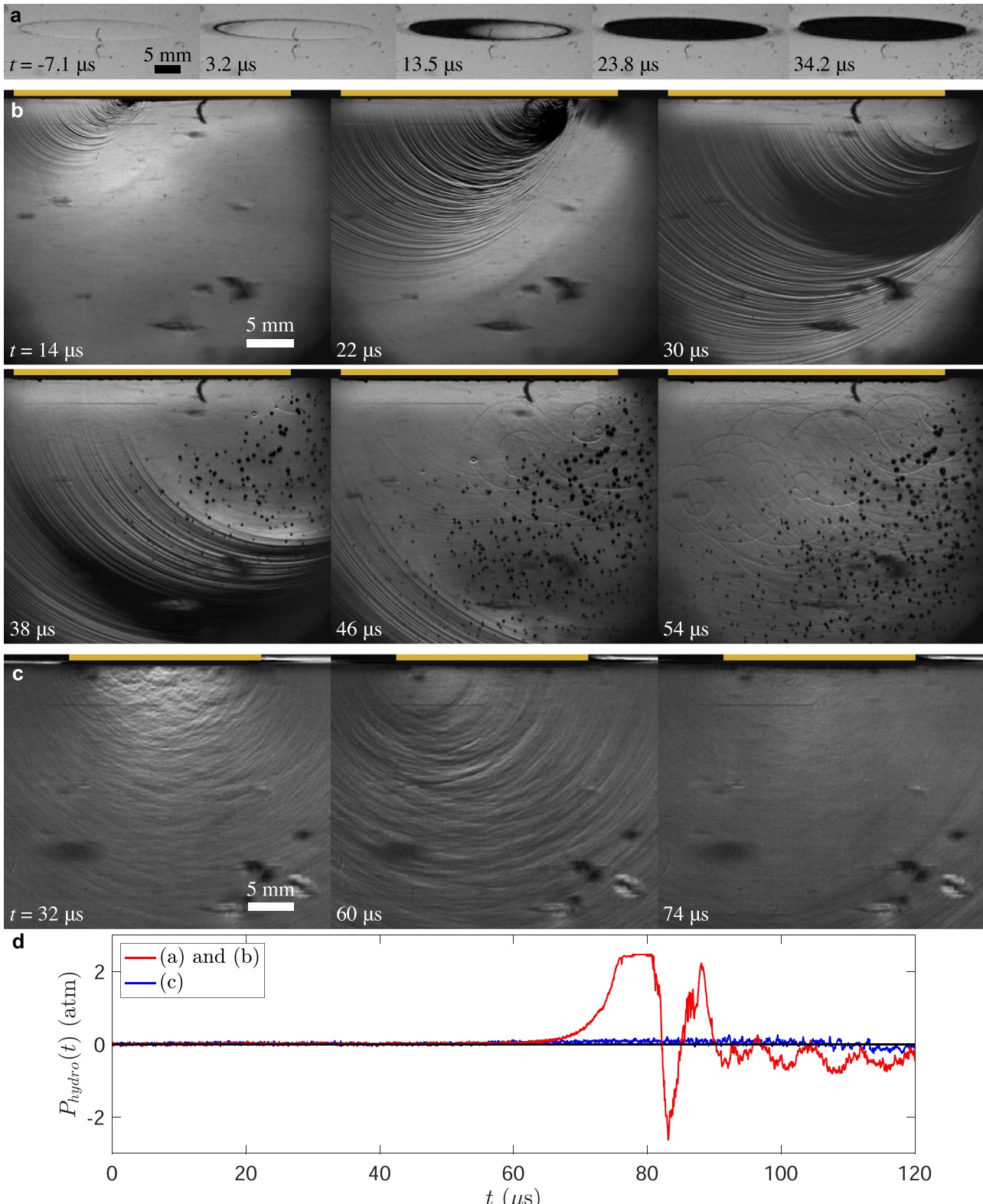

**Fig. 2 A cylinder impact onto a pool of water forms compression and tension waves. a** and **b** Show two views of a 30-mm-diameter cylinder forming cavitation bubbles in the liquid with an impact velocity of $U_o = 9.38$ m/s, an impact angle of $\alpha = 0.37°$, at $P_{amb} = 1$ atm. In **a** the view is slightly angled ( ~8°), looking up towards the free surface where the bottom of the cylinder comes in contact with the water at a slight angle from left to right. **b** A horizontal schlieren view of the same cylinder impact, showing the pressure waves also form from left to right. Light regions indicate rising pressure, and dark regions indicate decreasing pressure in the upward direction. **c** A similar impact of a 20-mm-diameter cylinder at $U_o = 9.14$ m/s, but larger $\alpha = 1.09°$, $P_{amb} = 1$ atm, where no cavitation occurs. The gold bar at the top of each frame indicates the cylinder's horizontal extent. The blotches are imperfections in the tank walls that are accentuated by the schlieren technique. A hydrophone signal of both impact events is shown in **d**, with recordings synchronized to the videos and $t = 0$ set to the initial water contact. The hydrophone resides approximately 96 mm away from the impact location causing a 65 µs delay, owing to the speed of sound. Supplemental movies 1–3 show **a**–**c**, respectively.

affects the air cushioning, which creates a second cavitation threshold. As the canonical cavitation number (1) does not predict cavitation onset well, nor these $P_{atm}$ effects, we propose a new formulation of the cavitation number to accurately predict cavitation onset in front of blunt bodies impacting on a liquid pool.

When the cylinder impacts the water, the pressure initially rises in the near field (Fig. 2b) by a value that scales with the water hammer pressure, shown theoretically[19] and confirmed experimentally[20] to be $\rho U_o c$. As the water hammer pressure is much greater than the ambient pressure, $\rho U_o c \gg P_{amb}$, the ambient pressure can be neglected and a background pressure of the order of the water hammer pressure develops under the cylinder as the new reference pressure:

$$P_r \sim \rho U_o c. \tag{2}$$

The pressure drop $P_d$ occurs when the high pressure on the bottom surface of the cylinder reaches the air-water interface at the edge of the cylinder and reflects with equal pressure amplitude, but opposite sign, because the acoustic impedance of water is much larger than air. Wagner type theories[2,21] describe the magnitude of the pressure on a solid surface impacting at angle $\alpha$ and show that the pressure on the submerged area is relatively constant, $\sim \rho U_o^2 \alpha^{-1}$, except near the contact line where a local maximum occurs. In the present experiments this local maximum is of such short duration ($O(1)$ ns) that upon reflection it does not cavitate the liquid as cavitation bubbles do not have sufficient time to form and grow ($O(100)$ ns is required[22–27]). Ignoring the short-duration local maximum, the effective pressure drop can be described as

$$P_d \sim \rho U_o^2 \alpha^{-1}. \tag{3}$$

For the present experiments $P_c \approx P_v \ll P_r$ and hence can be neglected. Combining (1), (2) and (3) yields a new cavitation number

$$Ca = \alpha \frac{c}{U_o} = \alpha Ma^{-1}, \tag{4}$$

where $Ma = U_o/c$ is the impact Mach number, and when $Ca$ is sufficiently small (i.e., $Ca < k_1$, where $k_1$ is a constant of order unity), cavitation is expected. Rearranging (4) shows that cavitation should occur when the impact angle is small enough:

$$\alpha < k_1 Ma, \tag{5}$$

and linearly bounded by the impact Mach number up to a constant scaling factor.

Eq. (5) predicts that cavitation can occur at any impact velocity as long as $\alpha$ is small enough. However, in the limit of $\alpha = 0$, the impact traps a thin air layer between the cylinder and pool, which cushions the impact and decreases the pressure on the bottom of the cylinder[28]. This cushioning effect can soften the impact so much that the schlieren imaging cannot detect the pressure waves for some impacts (see, for example, Supplemental Movie 4). Previous studies do not agree on the magnitude of the pressure on the impacting surface when air cushioning is significant[29–31], but we would expect the pressure to scale with $\rho U_o^2$[30], which upon reflection off the free surface results in

$$P_d \sim \rho U_o^2. \tag{6}$$

Daou et al.[20] experimentally showed that even in the presence of an air layer, the pressure in the near field continues to scale with the water hammer pressure (i.e., (2) still applies). Hence, combining (1), (2) and (6) and simplifying shows that even at the lowest $\alpha$ the impact must exceed a critical velocity for cavitation to occur;

$$Ma > k_2. \tag{7}$$

As the ambient gas pressure, $P_{amb}$, affects the amount of trapped air, and the cushioning effect, the value of $k_2$ should change with gas properties, decreasing with lower $P_{amb}$.

We test the validity of these theoretically predicted thresholds by performing an experiment in which we vary the impact velocity $U_o$, impact angle $\alpha$, ambient pressure $P_{amb}$, and cylinder diameter and plot the results of over two hundred cases in Fig. 3. The results show that (5) predicts cavitation well when $k_1 \approx 3$ (Fig. 3, dashed line). Therefore, for cavitation to occur, the Mach number of the contact line, $Ma_{cl} = U_{cl}/c$, must be greater than ~0.3, which is when compressibility effects of a fluid are expected to emerge. The second threshold, (7) also predicts the onset of cavitation. When $P_{amb} = 1$ atm, $k_2 \approx 0.003$ or $U_o \approx 4.5$ m/s in water (Fig. 3, solid line). When we decrease the ambient pressure the amount of air trapped between the cylinder and pool also decreases, which reduces the air cushioning effect (see Supplemental Movies 5 and 6). This allows cavitation to occur at lower impact velocities and $k_2$ decreases to $k_2 \approx 0.0022$ for $P_{amb} \leq 1/2$ atm, corresponding to $U_o \simeq 3.3$ m/s in water. Other studies have shown that the thinness of the air film and the rapid contact line motion cause the gas viscosity[31,32], gas density[32], rarefied gas effects[33], and gas speed of sound[29] to affect the air cushioning. Hence, we expect that the minimum velocity for cavitation and $k_2$ likely depend on each of these parameters as well. We further note that both thresholds (5) and (7) are independent of cylinder size for the range tested (diameters 10–40 mm) and (5) is independent of $P_{amb}$ as predicted by the theory. The number of bubbles that form behind the pressure wave increases with $Ma$ and cylinder diameter, reaching several hundred bubbles, which we roughly indicate in Fig. 3 (see legend) by counting the number of visible bubbles in the frame by eye.

Similar trends in the cavitation thresholds, (5) and (7), hold for impact on ethanol and a fully flourinated liquid called FC-72. Yet, the values of $k_1$ and $k_2$ differ, which we expect to be caused by the differing fluid properties such as the reduced ability to hold cavitation nuclei, which would alter $P_c$ in (1) (see Supplemental Information A).

We also find that the impacting body must contain an edge for cavitation to occur at the low impact velocities used in this study. In Fig. 4a we see the impact of an object with no edges, a 20-mm-diameter sphere, at 9.89 m/s. When the sphere impacts a weak compression wave propagates down into the pool, but we do not see a reflection forming a tension wave nor cavitation bubbles. If we add an edge by using a cylinder with a spherical cap on the front, we see how the pressure wave motion differs from a sphere and the essential influence of the edge for cavitation. Figure 4b and c show two synchronized views of a 50-mm-diameter cylinder with a bottom radius of curvature of 0.60 m, impacting at 7.75 m/s. First contact occurs in a ring around an air bubble caught at the lowest point[34] and then the contact line moves radially outward (Fig. 4b, $t = 7.4−49.1$ μs). Initially, the local $\alpha$ is small, so the contact line moves with supersonic velocity forming a compression wave with no reflections (Fig. 4c, $t = 8$ μs). With further submergence the local $\alpha$ increases, the contact-line velocity becomes subsonic and the waves overtake the contact line causing many small reflections and an oscillating pressure region ($t = 20$ μs). When the contact line reaches the edge, the wave reflects as a large tension wave (seen first on the right as thick dark lines at $t = 32$ μs then around the full circumference at $t = 42$ μs) which overlaps and sums to leave a cloud of cavitation bubbles in its wake ($t = 66−78$ μs). When the same curved cylinder impacts at lower velocity we see a similar sequence of events (Fig. 4d and e), except that by the time the contact line reaches the left and right edges (Fig. 4d, $t = 61.1$ μs) the pressure on the cylinder surface has decreased sufficiently that once again we do not see a reflection forming a large tension wave nor cavitation bubbles

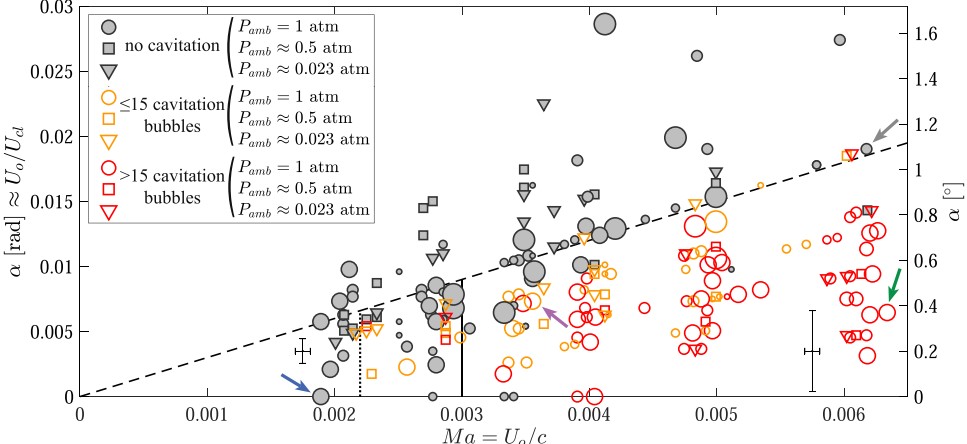

**Fig. 3 Plot showing cavitation onset when a flat-bottom cylinder impacts a pool of water.** Symbol explanations are shown in the legend and symbol size is proportional to the cylinder diameter, which ranges from 10 to 40 mm. The dashed line plots (5) with $k_1 = 3$, the solid line plots (7) with $k_2 = 0.003$ (for $P_{amb} = 1$ atm), and the dotted line plots (7) with $k_2 = 0.0022$ (for $P_{amb} \leq 1/2$ atm). The green and gray arrows indicate the cases shown in Fig. 2b and c respectively and the blue arrow indicates the case shown in Supplemental Movie 4, in which the pressure waves are so weak the schlieren imaging cannot detect them. Each marker represents one impact event. The number of cavitation bubbles varied greatly with 779 bubbles counted for the case indicated by the green arrow and only 6 counted for the case indicated by the purple arrow. The bubble quantity is roughly indicated by the symbol color; see legend. Typical 95%-confidence-uncertainty bands are shown on the left and right of the plot for a 20-mm-diameter cylinder with the uncertainty on $\alpha$ increasing linearly with $U_o$ between them.

(Fig. 4e, $t = 64\,\mu s$). Hence, not only is an edge required for cavitation, but it must reside at a position where the reflection can decrease the pressure sufficiently to cavitate the liquid.

The spherical geometry changes some aspects of the impact event altering the cavitation and its prediction. As the pressure waves originate in the center and travel radially outward, the summing and focusing of the waves differs causing cavitation to initiate deeper in the pool (Fig. 4c, $t = 66\,\mu s$) instead of directly beneath the last edge to impact (Fig. 2b, $t = 30\,\mu s$). The curvature also changes the shape of the free surface depression[31] such that assuming a flat free surface no longer appears sufficient to approximate $\alpha$. As these effects complicate the reflected pressure and the pressure field, additional modeling will be required to predicted the onset of cavitation for such variations in geometry.

These results provide experimental evidence and scaling laws showing that cavitation occurs in solid-liquid impacts more often than previously thought, where impact speeds are above 4.5 m/s, impact angles are less than ~1°, and the impacting body has an edge where a reflection can occur.

## Methods

Supplemental Fig. 1 shows a schematic of the setup. Flat-ended and rounded steel cylinders were dropped from various heights up to 4.75 m onto a pool of water ($\rho = 1000$ kg/m³, $c = 1480$ m/s, $P_v = 2.3$ kPa), ethanol ($\rho = 800$ kg/m³, $c = 1160$ m/s, $P_v = 6.0$ kPa), or 3M™ Fluorinert™ Electronic Liquid FC-72 ($\rho = 1680$ kg/m³, $c \approx 520$ m/s, $P_v = 30.9$ kPa). Drops occurred inside a vacuum chamber in which we varied the pressure from the vapor pressure to 1 atm (only varied for water). The flat-ended cylinder diameters varied from 10–40 mm with the length held constant at 100 mm. The roughness of the impacting surface of a sample cylinder was measured with a Dektak 150 surface profiler, which gave an average roughness $R_a = 91$ nm, an RMS roughness $R_q = 114$ nm, and the total roughness (i.e., maximum valley to peak distance) $R_t = 784$ nm. The back end of the cylinder had a string attached on the axis, which extended up and over two pulleys and then attached to a small counterweight. A solenoid held the string and cylinder in place and when released the cylinder dropped. The string helped the cylinder to impact at small angles $\alpha$, which varied randomly. If desired the string could be adjusted slightly off axis to increase $\alpha$.

The impact event was recorded with two cameras and two hydrophones. A Kirana-5M high-speed video camera, controlled with Kirana control software, from Specialized Imaging recorded the impact events at up to 5 million frames per second (fps) through a basic lens schlieren system, with a spatial resolution of 32–59 μm per pixel. The schlieren system consisted of a pulsed diode laser (10 ns pulse duration, SILUX-640, Specialised Imaging) that passed through a slit

aperture, the first field lens (75 mm diameter, 500 mm focal length, planoconvex), the test section including the vacuum chamber and tank, the second field lens (75 mm diameter, 500 mm focal length, planoconvex), a horizontal knife edge (covers the bottom rays), focusing lens (50 mm diameter, varied between 200–400 mm focal length, biconvex) and into the camera with no lens attached. The horizontal knife edge covering the bottom rays produced images in which light regions indicate where the pressure increases and dark regions indicate where the pressure decreases in the upward direction. The light intensity of each region is an integration of the pressure gradient in the out-of-page direction and the depth of field is approximately equal to the width of the tank, due to the collimation of the light. A Phantom v2511 high-speed camera controlled by PCC software recorded at up to 99 kfps with a spatial resolution of 88–121 μm per pixel. It sat slightly below the free surface level, looking up at the impact location at approximately 8°. Two calibrated Müller-Platte Needle Probe hydrophones from Müller Instruments with a rise time of 40 ns were placed 78 mm below the water surface on opposite sides of the impact directed up towards the impact location. The distance from the center of the cylinder at the free surface to each hydrophone was 96 mm. The hydrophones connected to a Tektronix DPO7254 oscilloscope that recorded pressure readings at 20 MHz and has a rise time of 160 ps. When the cylinder approached the tank of water an optical trigger sent a signal to the two cameras and oscilloscope to start and synchronize the video and audio recordings. We measured the impact velocity $U_o$ and angle $\alpha$ from the Phantom camera. The Kirana camera let us visualize the motion of the pressure waves, and both cameras were used to detect cavitation. Some of the images in the figures have been brightened using Adobe Lightroom. Data analysis and plotting was accomplished in MATLAB 2019.

As the impact angle $\alpha$ is important in determining the occurrence of cavitation, we detail how it was measured and the uncertainty in this measurement. The impact angle was calculated using the small angle approximation as $\alpha = U_o \Delta t/d$, where $\Delta t$ is the time between first contact of the cylinder and full submergence of its bottom surface and $d$ is the cylinder diameter. We assume that the free surface of the pool is flat. We find this to be a good assumption, despite the small surface depression caused by the approaching cylinder, by comparing the calculated contact-line velocity $U_{cl} = U_o/\alpha = d/\Delta t$ to the measured velocity of the origin of the pressure waves seen in the schlieren images for specific cases. The majority of the uncertainty in $\alpha$ stems from the uncertainty of $\Delta t$ and is caused by the limited time resolution of the Phantom camera, 10 μs. The zeroth order uncertainty of finding the initial-contact and full-submergence frames is ±0.5 frame, which results in a 95% confidence uncertainty on $\Delta t$ of $\epsilon_{\Delta t} = 7$ μs. The uncertainties of the other variables are small enough that they can be neglected by the quarter rule for most of the data. Using the Taylor series method, the uncertainty on $\alpha$ is $\epsilon_\alpha = U_o \epsilon_{\Delta t}/d$. Figure 3 shows two representative values of the 95%-confidence-uncertainty at low and high $U_o$ for the most common cylinder diameter used, $d = 20$ mm. $\epsilon_\alpha$ increases linearly between them. A similar method for calculating the local impact angle on convex-bottomed cylinders will not produce as accurate results as the assumption that the water surface is flat is less valid. Hence, both the cylinder and pool surface have an angle that must be taken into account to find the local impact angle which changes with radial position. The Taylor series method was also applied for the 95% confidence uncertainty on $Ma$ which is also shown in Fig. 3.

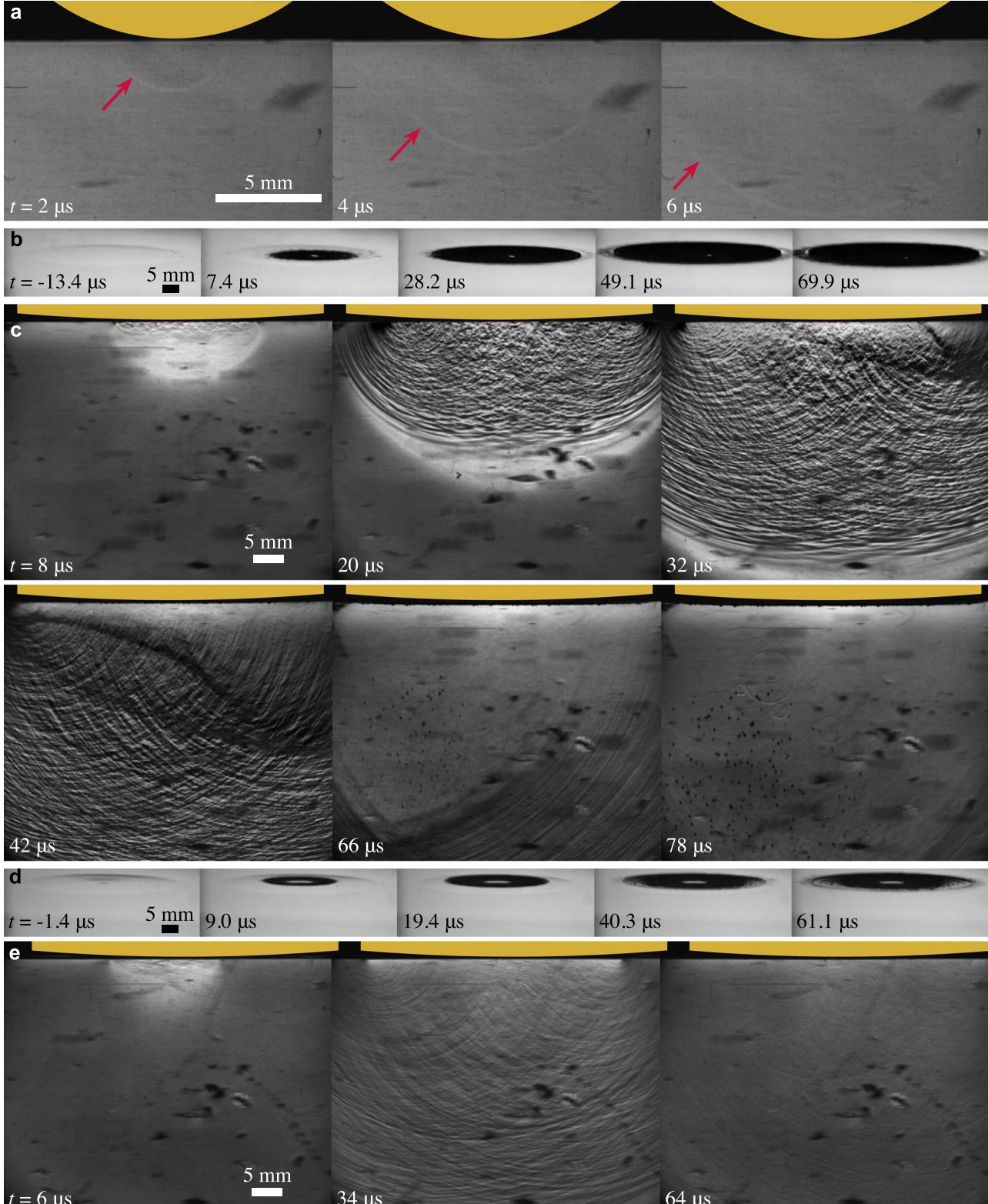

**Fig. 4 The impacting body must have an edge for cavitation to occur. a** The impact of a 20-mm-diameter sphere at 9.89 m/s forming a weak compression wave (red arrows) that emanates into the pool without causing cavitation. **b** and **c** Show two views of a 50-mm-diameter cylinder with a convex curvature of radius 0.60 m on the bottom surface impacting at 7.75 m/s. The tension wave that reflects when the edge submerges ((**c**) $t = 32-42\,\mu s$) causes cavitation where the waves overlap. **d** and **e** Show two views of the same curved cylinder impacting at 3.47 m/s. By the time the edges submerge the pressure amplitude near the contact line has decreased sufficient such that a reflected tension wave (thick, dark region) is not visible and no cavitation occurs. The gold shapes at the top of each frame indicate the approximate position, shape, and size of the lower portion of the impacting body. **a**, **c** and **e** are schlieren images with light regions indicating increasing pressure in the upward direction and dark regions indicating decreasing pressure. The ambient pressure in all cases is 1 atm. Supplemental movies 7–11 show **a–e**, respectively.

## Data availability

Source data are available for this study. The data used to generate Fig. 2d and Fig. 3, and Supplementary Fig. 2 can be found in the Supplementary Data 1 and 2 respectively. All other data that support the plots within this paper and other findings of this study are available from the corresponding author upon reasonable request.

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

## Acknowledgements

S.T.T. acknowledges early discussions with T. G. Etoh and K. Takehara. This work was supported by King Abdullah University of Science and Technology (KAUST).

## Author contributions

S.T.T. conceived the research and took the initial data; N.B.S., K.R.L. and T.T.T. designed the research; N.B.S. and K.R.L. took the data; N.B.S., K.R.L. and Z.P. analyzed the data; N.B.S. and Z.P. wrote the paper; and all authors edited the paper.

## Competing interests

The authors declare no competing interests.
