## [Transparent Peer Review File · Nature Communications]

Peer Review Information

Manuscript title: Cavitation upon low-speed solid-liquid impact

Corresponding author name(s): Nathan Speirs

Reviewer comments & decisions:

Reviewer comments, first version:

Reviewer #1 (Remarks to the Author: Overall significance):

The manuscript "Cavitation in the early moments of low-speed solid-liquid impact" shows and justifies the existence of cavitation at much lower velocity for the first time. New cavitation number is proposed for the prediction. The importance of the impact angle is clearly demonstrated for the cylinder impact to the liquid pool. The research, of course, is of general interest of broad range of readers including industry.

The authors appropriately shortly cited previous works. And the findings are discussed in the context of previous literature.

Reviewer #1 (Remarks to the Author: Impact):

Nature Physics is most relevant journal. I feel that new series of experiments of different configuration would appear in the next future. So that short time cavitation can be detected with the help of high-speed recordings, what became possible only now.

Reviewer #1 (Remarks to the Author: Strength of the claims):

1. There is some need to have some simple correlation curve to have some generalization of cavitation occurrence.
2. How will the gas phase influence on the cavitation? Authors can evacuate the air and fill the chamber by another gas.

3. Will be there some dependence on the density relation?
4. It would be useful to have information about steel surface roughness, which is contacting with liquid.
5. Some more discussion for the cylinder edge influence is needed. What will happen for a truncated cone? or cylinder with chisels?
6. It seems that for low pressure the cavitation is most likely to exist at wider range of Mach number. Please, discuss this aspect.
7. Why authors have not provided the data for the biggest cylinders at low pressure?

Reviewer #1 (Remarks to the Author: Reproducibility):

1. Angle (α) definition is most important thing since this parameter influence on the criteria. Authors should write in detail how it should be determined correctly.
2. Error in angle determination should be specified. As well as resolution of the system providing the data on angle should be provided. Schlieren / shadow pictures resolution, for example.

Reviewer #2 (Remarks to the Author: Overall significance):

I read the manuscript with immense interest. The authors are trying to address a classical problem with a new perspective with the aid of more advanced instrumentations.

While I agree that paper contains a lot of in-depth views on the impact phenomenon and the authors have proposed an modified chart to show their significance, I am not sure whether the number of bubbles created during the impact did actually correlate with the conclusions. Even if it does, it may be even better to provide a short descriptions on the experimental errors for those results obtained from the video, especially on the bubble counting methods and the estimation on the propagation of waves. The video clips only show a 2-D view of the impact flow field and the depth of the field was not mentioned.

A minor comment will be on the reference on the cliff climbing in the first paragraph. I am not sure whether it bears any significance to the overall impact of the paper.

Reviewer #2 (Remarks to the Author: Impact):

The paper is nicely written and properly executed experimental paper but I couldn't find the assessment on the experimental errors anywhere. This is may be the set back I have for the paper.

Reviewer #2 (Remarks to the Author: Strength of the claims):

The results together with the video support the results obtained.

Reviewer #2 (Remarks to the Author: Reproducibility):

N.A.

Reviewer #3 (Remarks to the Author: Overall significance):

This is an engaging article which presents an interesting new examination of the cavitation that can occur when a blunt solid impacts a liquid surface. While the presence of cavitation due to a water hammer like effect is known for a liquid striking a solid, here the reverse is examined. In the present paper, it is argued using both precise experimentation and scaling analysis, that low-speed impacts initiate cavitation through the formation of a negative pressure region. This region is brought about by a reflected compression wave formed at the moving contact line of the angled impacting cylinder. It is further shown that the classical cavitation number is not able to predict the onset of cavitation in this system, and thus a new cavitation number is required. The paper formulates this cavitation number as a balance between the high pressure generated by the impacting base of the cylinder and the dynamic pressure drop when said pressure wave reaches the edge of the cylinder. The argument for a compressibility driven mechanism is strengthened by experimental data showing that cavitation occurs when the Mach number of the contact line is greater than ~ 0.3 . The paper characterizes a novel system, where geometry (angle of impact) enhances the dynamics to bring about cavitation in an unexpected scenario.

Reviewer #3 (Remarks to the Author: Impact):

I believe that this work expands on where the community will look for cavitation in the future. Given the potential for cavitating bubbles to cause damage, it is important to highlight and understand when they show up in unexpected systems.

Reviewer #3 (Remarks to the Author: Strength of the claims):

The work is convincing, however I have some concerns, which are listed below.

Major:

1. The paper neglects to mention cavitation at low velocities in the analogous system of a fluid being

accelerated relative to a solid, the converse of the system examined in this paper. The authors' own previous work ("Cavitation onset caused by acceleration", PNAS 2017) recognizes that the classical cavitation number does not apply in systems with cavitation brought on by relatively low maximum impact velocities (~ 2 m/s) and thus constructs a novel cavitation number to describe the onset of cavitation. How is the current work fundamentally different? Bridging the gap between the converse system and the one examined in the current paper would put the current work in a better context and help the reader assess the extent to which the authors are describing a new geometry as opposed to a new mechanism.

2. Equation (7) is presented as a simplified combination of equations (1), (2), and (6). However, as written, does it only hold for the case of $n = 2$? Given the dependence of n on α , at low values of α , it appears that the threshold given by equation (7) will not accurately capture the onset of cavitation at low values of α . From the experimental data in figure 2, this threshold for ($k_2 = 0.003$) appears murky, and stands in stark contrast to the clear threshold given by equation (5) (for $k_1 = 3$). The supplemental data also does not provide strong evidence for this Mach number threshold as decreasing k_1 separately for each of the two fluids would describe most of the "no cavitation" points shown. It appears the threshold given by equation (7) may be too simplistic to describe the behavior observed in this system. Can the authors clarify the n dependence and its relationship to the experimental data?

3. The constant k_2 in equation (7) is said to decrease with the ambient pressure. It would be good to comment, based on the values used in this study, how much of an effect the reductions in pressure would have on the constant. It appears that in only the highest ambient pressure studied is cavitation clearly arrested at lower Mach numbers.

Minor:

1. Figure 1 is quite a striking high contrast image. That said, I recommend, in the caption, orienting the reader to the direction of travel of the cylinder prior to impact.

2. In figure 2d the pressure for the hydrophone appears to drift negatively with time for (a) and (b). Is this drift from the measuring equipment, or was the bath not initially quiescent?

3. The sentence "This large tension wave initiates cavitation just beneath the right edge of the cylinder where last contact occurs (Fig. 2a at $t = 34.2 \mu\text{s}$)" I believe is incorrect. Last contact based on figure 2b appears to be earlier than $30 \mu\text{s}$. Based on the position of the wave propagation front in fig. 2b at $22 \mu\text{s}$ it appears to have ~ 5.6 mm to reach the right edge, which given the contact line speed of ~ 1455 m/s would take $\sim 3.8 \mu\text{s}$. Was the intent to reference (Fig. 2a at $t = 23.8 \mu\text{s}$) here?

4. The squared velocity term in equation (3) includes both the contact line and impact velocities. Can the

authors comment on this choice?

Reviewer #3 (Remarks to the Author: Reproducibility):

Given the level of detail provided in the manuscript and supplementary material, I believe the results of this work can be reproduced by another researcher. Overall, I find that the experimental data is presented in a clear and consistent manner.

Author rebuttal, first version:

**Responses to Reviewer and Editor comments for submission 1
for
Cavitation in the early moments of low-speed solid-liquid impact
by
Nathan Speirs, Kenneth Langley, Zhao Pan, Tadd Truscott, and Sigurdur Thoroddsen**

Reviewer 1

Overview

The manuscript "Cavitation in the early moments of low-speed solid-liquid impact" shows and justifies the existence of cavitation at much lower velocity for the first time. New cavitation number is proposed for the prediction. The importance of the impact angle is clearly demonstrated for the cylinder impact to the liquid pool. The research, of course, is of general interest of broad range of readers including industry. The authors appropriately shortly cited previous works. And the findings are discussed in the context of previous literature.

Nature Physics is most relevant journal. I feel that new series of experiments of different configuration would appear in the next future. So that short time cavitation can be detected with the help of high-speed recordings, what became possible only now.

1.1 There is some need to have some simple correlation curve to have some generalization of cavitation occurrence.

Response:

We strongly agree with the reviewer that there needs to be some generalization of cavitation occurrence caused by the large pressure waves generated in solid-liquid impact events. The related literature cited in the introductory paragraphs of the manuscript indicate that this type of cavitation is possible yet they do not discuss specific threshold criteria. The plot in Fig. 3 helps to fill this important gap in the literature and includes equations 5 (dashed line) and 7 (solid line) from our theory, which separate the data well.

1.2. How will the gas phase influence on the cavitation? Authors can evacuate the air and fill the chamber by another gas.

Response:

The gas should only affect the amount of air cushioning and hence the threshold Mach number that must be exceeded for cavitation to occur. We have examined this in our study by changing the ambient air pressure and hence the air density in the vacuum chamber and have found that at standard atmospheric conditions the critical Mach number is $Ma \approx 0.003$. When the air pressure is decreased to $P_{amb} \leq 0.5$ atm the density decreases also and the critical Mach number decreases to $Ma \approx 0.0022$. Using a different gas as the reviewer recommends, would indeed be interesting as it would decouple the gas density from the pressure, but unfortunately is impractical and

unsafe for our set up. This is because for each cylinder drop we have to open the vacuum chamber to reset the cylinder. As the chamber is large this would require a large amount of gas to be expelled for each drop which would enter the small room with relatively little ventilation. Yet, a related work on droplet impacts onto a solid surface by Burzynski & Bansmer (*Phys. Rev. Fluids*, 2019) found that changing the gas density and viscosity, by using different gases, can lead to a larger entrapped bubble, which they found by measuring the radial extent of the entrapped air disc. Hence, we would expect that if we could fill our chamber with a higher density gas such as SF₆ at ambient pressure, the effect of the gas cushioning would increase, decreasing the amplitude of the pressure waves that the liquid experiences and pushing the threshold Mach number to a higher value.

1.3. Will be there some dependence on the density relation?

Response:

Please see our response to your previous comment, where we have addressed this question.

Editor Comment on 1.

Additional experiments are required here.

Response:

As mentioned above in response to comment 1.2 we have already investigated the effect of air density on the air cushioning and the cavitation phenomena by changing the ambient air pressure, which also changes the air density over a large range. We show and discuss its effect in the paper and in response to the above reviewer comment. Although using different gases in the chamber would be interesting as it would decouple the gas pressure and density it is unsafe for our experimental setup and related works found in the literature already discuss how changing the gas density affects air cushion. We used this to explain the expected result of changing gas density on cavitation.

2.4. It would be useful to have information about steel surface roughness, which is contacting with liquid.

Response:

We measured the surface roughness on the bottom of a representative cylinder using a Dektak 150 surface profiler. The average roughness is $R_a = 91.04$ nm, the RMS roughness is $R_q = 114.03$ nm, and the total roughness (i.e., maximum valley to peak distance) is $R_t = 784.27$ nm. This information has now been added to the first paragraph of the Methods section of the paper.

2.5. Some more discussion for the cylinder edge influence is needed. What will happen for a truncated cone? or cylinder with chisels?

Response:

We thank the reviewer for this comment as we neglected to directly mention the important influence of the edge in the previous version of the paper. As the reviewer suggests, the influence of the edge is vitally important for cavitation to occur at such low impact velocities. We now report on additional experiments with spheres and cylinders with a curved bottom face to emphasize the important influence of the edge and where it must be located. Parts of this discussion were previously in Supplemental Information B, but this has been moved to the main paper and can now be found in the 3rd to last paragraph before the Methods section and in the new Fig. 4.

Editor's Comment on 2.

The angle α is a rather special parameter in the experimental setup and the theoretical analysis; what would happen, e.g., if a spherical body would be used instead? A cylinder with a curved bottom face?

This is to our view an important question.

Response:

We now report on the impact of a sphere and cylinders with a curved bottom face in the main body of the paper. We do not see any cavitation for sphere impact, but do see cavitation for the impact of cylinders with a curved bottom face. These experiments show that the presence of an edge is vitally important for cavitation to occur at such low impact velocities, as the reviewer suggests. The curved bottom face also changes other aspects of the impact events, such as the motion of the pressure waves, and location of cavitation. Yet the same basic phenomena are at play; high pressure generated at impact with a large wave reflected at the cylinder edge that creates a negative pressure region and cavitation. These details are now discussed in the 2nd-to-last and 3rd-to-last paragraphs before the Methods section. With a curved bottom surface the local angle α changes as the cylinder submerges causing the local pressure on the surface to change with submergence (ref. [2]). Measuring α for such cases is difficult as the free-surface depression seen before the cylinder contacts the water is now more curved so that both the cylinder and pool surface have an angle that must be taken into account to find the local α and hence the reflected pressure. Hence, predicting cavitation for curved bottom faces is more complicated and an improved understanding of the shape of the free-surface depression is needed. This is now briefly discussed in the last paragraph of the Methods section.

3.6. It seems that for low pressure the cavitation is most likely to exist at wider range of Mach number. Please, discuss this aspect.

Response:

Yes, the lower ambient pressures allow for cavitation to occur at lower impact Mach numbers. This occurs because at low impact angles the cylinder traps a small volume of air between itself and the pool that acts to cushion the impact and reduce the pressure on the bottom surface of the cylinder. Hence, a lower amplitude pressure wave reflects off of the air interface. When the ambient gas pressure decreases, the cylinder traps less air. This decreases the cushioning effect and generates a higher pressure on the cylinder surface which reflects and allows cavitation at

lower impact Mach numbers. We have now altered the paper to indicate this more clearly, showing that the lower limit for cavitation is $Ma \approx 0.003$ at normal atmospheric conditions and $Ma \approx 0.0022$ when $P_{amb} \leq 0.5$ atm. This effect of the ambient pressure is now introduced before the discussion of the mathematical model in the first paragraph in the right hand column of pg. 3 of the paper. We have also added a vertical dotted line to Fig. 3 at $Ma = 0.0022$. The discussion in the paragraph after Eq. 7 (last paragraph on pg. 4) has also been altered to better reflect the effect of the ambient pressure and to discuss other fluid properties that would likely affect the threshold Mach number.

3.7. Why authors have not provided the data for the biggest cylinders at low pressure?

Response:

We only used the 20-mm-diameter cylinder under vacuum pressures for a few reasons. First, since there was little effect from the diameter at atmospheric pressure, we do not anticipate any large effect of changing the diameter under reduced pressure. Second, the vacuum experiments were time consuming as the chamber must be cleaned, sealed, evacuated, and refilled for each test, so getting repeated data for different cylinder sizes would have added at least a few weeks to the data collection. Third, the larger cylinders have a higher risk of damaging the setup (i.e. chipping the tank and breaking the hydrophones), and so we wanted to minimize the number of drops of the large diameter cylinders. Experiments with the largest cylinder under vacuum are therefore not practical.

Editor's comment on 3.

Additional data analysis and discussion is required here.

Response:

We have improved the discussion on why cavitation is seen at lower Mach numbers for the lower ambient pressure impacts in the paragraph that spans pg. 4-5 of the paper. We have also added a second vertical line to Fig. 3 to help make the effect of the ambient pressure more clear.

Most important issues to the reviewer:

4.1. Angle (α) definition is most important thing since this parameter influence on the criteria. Authors should write in detail how it should be determined correctly.

Response:

We have now added a paragraph at the end of the Methods section describing how alpha was determined by measuring the time between first contact of the cylinder with the pool and full submergence of its bottom surface. Using this time along with the cylinder diameter d and impact velocity U_o allows us to calculate the impact angle with the equation $\alpha = U_o \Delta t / d$ using the small angle approximation.

4.2. Error in angle determination should be specified. As well as resolution of the system providing the data on angle should be provided. Schlieren / shadow pictures resolution, for example.

Response:

We have added a discussion on the uncertainty of the determination of α in the last paragraph of the Methods section. As stated in that paragraph this uncertainty mainly stems from the limited temporal resolution of the Phantom camera, which is 10 μ s. We convert this to an angle uncertainty using the Taylor series method, which increases linearly with the impact velocity. Fig. 3 shows 95%-confidence-uncertainty bands at the left and right sides of the plot to show how the uncertainty changes with U_o or Ma . The uncertainty in α increases linearly between these two points.

We have also added the spatial resolution for both high-speed cameras in the second paragraph of the Methods section.

Editor's comment on 4

This is in line with our assessment, compare comment 2 above and the report of reviewer #2 below.

Response:

We have added additional description on how the angle is determined and the uncertainty in its measurement in the last paragraph of the Methods section. Supplemental Fig. 1c also shows a sketch of how the alpha angle is defined.

Reviewer 2

I read the manuscript with immense interest. The authors are trying to address a classical problem with a new perspective with the aid of more advanced instrumentations.

Overview:

While I agree that paper contains a lot of in-depth views on the impact phenomenon and the authors have proposed an modified chart to show their significance, I am not sure whether the number of bubbles created during the impact did actually correlate with the conclusions. Even if it does, it may be even better to provide a short descriptions on the experimental errors for those results obtained from the video, especially on the bubble counting methods and the estimation on the propagation of waves. The video clips only show a 2-D view of the impact flow field and the depth of the field was not mentioned.

Response:

We thank the reviewer for the positive feedback on our paper and are pleased that you found it so interesting! We now provide a description of the experimental uncertainty on the relevant measured and calculated quantities in the Methods section of the paper and in Fig. 3 per your request and further address the experimental uncertainty in response to your comment below.

The main purpose of the study was to discover and describe when cavitation occurs in the early stage of water impact. Hence, in general, we did not count the precise number of cavitation bubbles for every impact. But, from watching the numerous videos we noted that the number of bubbles varied greatly with 779 bubbles counted in frame for the case indicated by the green arrow on the far right of the plot in Fig. 3 while only 6 bubbles were seen for the case indicated by the purple arrow toward the middle of the plot. Finding the exact number of bubbles for each impact would have required more high-speed cameras as part of the bubble clouds often left the frame of the cameras. We roughly indicate the number of bubbles counted in frame as being greater or less than 15, which was arbitrarily chosen to provide the reader a general sense of the increase both with cylinder size and the Mach number. This as well as the two cases mentioned above is now stated in the paragraph after Eq. 7 and in the caption of Fig. 3.

The depth of field of the schlieren system is now stated in the Methods section where we indicate that it is approximately equal to the width of the water tank as the light between the two field lenses is collimated. We have also noted in the Methods section that the light intensity in the schlieren images is affected by the integration of the pressure gradient in the out-of-page direction and hence cannot be thought of as a cross-section of the pressure field.

A minor comment will be on the reference on the cliff climbing in the first paragraph. I am not sure whether it bears any significance to the overall impact of the paper.

Response:

The cliff jumping reference is used to increase general interest in the paper and it may be possible under certain circumstances for cavitation to occur in such impacts; particularly if the jumper wore shoes that impacted very flat.

The results together with the video support the results obtained.

1. The paper is nicely written and properly executed experimental paper but I couldn't find the assessment on the experimental errors anywhere. This is may be the setback I have for the paper.

Response:

We have now added a paragraph at the end of the Methods section describing how we calculate the experimental uncertainty for the plot shown in Fig. 3. This was done with the Taylor series method. The largest uncertainty is in the determination of the impact angle α , which is due to the limited temporal resolution of the Phantom camera. The uncertainty on α and Ma have been quantified as described in the Methods section and typical 95%-confidence-uncertainty bands are shown in Fig. 3, which shows how the uncertainty on α increases linearly with the impact velocity U_o . Each data point shown in Fig. 3 represents one impact event. The caption of Fig. 3 also has an improved explanation of the uncertainty bands shown.

Editor comment on this.

The prototypical error bar given in Figure 3 is not enough. Please provide #repeats per data point, e.g., as Supplementary Information. Please also see the requests of reviewer #1 regarding the measurement of alpha above.

Response:

We have now provided a much more expanded description of how we calculate the measurement uncertainty of the data in the Method section of the paper. We describe the uncertainty in α , the impact velocity U_o or Mach number Ma and state the image resolutions, time resolution of the high-speed cameras, and the rise times of the components of the hydrophone system. In the caption of Fig. 3 we now state that “Each marker represents one impact event,” and that the “95%-confidence-uncertainty bands are shown on the left and right of the plot for a 20-mm-diameter cylinder with the uncertainty on α increasing linearly with U_o between them.” Note, we provide rigorous uncertainty quantification using the classic method based on Taylor expansion for each data point. This is why the uncertainty on α increases linearly with the impact velocity U_o and it is unnecessary to clutter the plot with uncertainty bands on every data point.

Reviewer 3

Overview:

This is an engaging article which presents an interesting new examination of the cavitation that can occur when a blunt solid impacts a liquid surface. While the presence of cavitation due to a water hammer like effect is known for a liquid striking a solid, here the reverse is examined. In the present paper, it is argued using both precise experimentation and scaling analysis, that low-speed impacts initiate cavitation through the formation of a negative pressure region. This region is brought about by a reflected compression wave formed at the moving contact line of the angled impacting cylinder. It is further shown that the classical cavitation number is not able to predict the onset of cavitation in this system, and thus a new cavitation number is required. The paper formulates this cavitation number as a balance between the high pressure generated by the impacting base of the cylinder and the dynamic pressure drop when said pressure wave reaches the edge of the cylinder. The argument for a compressibility driven mechanism is strengthened by experimental data showing that cavitation occurs when the Mach number of the contact line is greater than ~ 0.3 . The paper characterizes a novel system, where geometry (angle of impact) enhances the dynamics to bring about cavitation in an unexpected scenario.

I believe that this work expands on where the community will look for cavitation in the future. Given the potential for cavitating bubbles to cause damage, it is important to highlight and understand when they show up in unexpected systems.

1.1. The paper neglects to mention cavitation at low velocities in the analogous system of a fluid being accelerated relative to a solid, the converse of the system examined in this paper. The authors' own previous work ("Cavitation onset caused by acceleration", PNAS 2017) recognizes that the classical cavitation number does not apply in systems with cavitation brought on by relatively low maximum impact velocities (~ 2 m/s) and thus constructs a novel cavitation number to describe the onset of cavitation. How is the current work fundamentally different? Bridging the gap between the converse system and the one examined in the current paper would put the current work in a better context and help the reader assess the extent to which the authors are describing a new geometry as opposed to a new mechanism.

Response:

This is a very interesting question and we can see the similarity between the two systems that you suggest as the impact of an object on a pool of water does involve large accelerations as shown in many previous works, including several of our own. Yet, despite the similarity, the cavitation reported in the current study is an acoustic phenomenon caused by the compressibility of the water upon impact, which forms pressure waves that reflect and create low pressure zones. The cavitation that occurs at low velocity discussed in our PNAS paper results from the acceleration of an incompressible liquid as described by the incompressible Navier-Stokes equation. This fundamental difference is evidenced by several features of the impact events as listed here.

- In the current study cavitation is not seen until the contact line velocity reaches approximately $\frac{1}{3}$ the speed of sound in the liquid, which is when compressibility effects are expected to emerge. No such mach number dependence (contact line speed or impact angle dependence) was observed in the PNAS paper, but rather an acceleration dependence.
- In the current study cavitation occurs behind the front of the expansion wave, which is off of the body, but it was at the wall of the container in the PNAS paper
- In the current study the ambient pressure is not a significant factor for the onset of cavitation (at higher velocities) because a compression wave first raises the local pressure before the tension wave decreases the pressure. For the acceleration-induced cavitation in the PNAS paper, the onset of cavitation was found to depend on the ambient pressure showing that such wave motion was not the key factor for the onset of cavitation.
- For acceleration-induced cavitation, the characteristic length of fluid that accelerates is critical for determining the pressure drop and hence cavitation. In the current study we do not find any such length scale that determines the pressure drop and cavitation. The depth of the liquid in the tank appeared to be insignificant as cavitation occurred before the pressure waves reached the bottom.

We now mention the difference between the two types of cavitation in the 3rd paragraph of the paper where we say the following.

“The sudden impact compresses the liquid beneath, and the resulting pressure waves emanate outward at the speed of sound in the liquid ($c = 1480$ m/s in water) forming a cloud of cavitation bubbles in the wake of the negative pressure wave. This type of cavitation is caused by liquid compression and pressure wave reflection and hence is fundamentally different from acceleration-induced cavitation which is also seen at low flow velocities [13–15].”

Editor’s comment on 1.

This is a serious concern and must convincingly be address if publication in Nature Communications is intended.

Response:

The cavitation in this study is caused by the compressibility of the water upon impact, which forms pressure waves that reflect and create low pressure zones. This is fundamentally different from the acceleration-induced cavitation discussed in our previous PNAS paper which can be predicted from the incompressible Navier-Stokes equation with no reflected pressure wave required in the theory. The differences are evidenced by the several key features detailed in response to the reviewer comment listed above. We have now cited our PNAS paper in the introduction of the paper.

2.2. Equation (7) is presented as a simplified combination of equations (1), (2), and (6). However, as written, does it only hold for the case of $n = 2$? Given the dependence of n on α , at low values of α , it appears that the threshold given by equation (7) will not accurately capture the onset of cavitation at low values of α . From the experimental data in figure 2 (do you mean Fig.

3?), this threshold for ($k_2 = 0.003$) appears murky, and stands in stark contrast to the clear threshold given by equation (5) (for $k_1 = 3$). The supplemental data also does not provide strong evidence for this Mach number threshold as decreasing k_1 separately for each of the two fluids would describe most of the “no cavitation” points shown. It appears the threshold given by equation (7) may be too simplistic to describe the behavior observed in this system. Can the authors clarify the n dependence and its relationship to the experimental data?

Response:

The pressure on the bottom of a body impacting on a pool with an intervening air layer is a complicated problem that has not been fully resolved in the literature and the various studies on the topic [29-31] disagree on an appropriate expression for this pressure. Equation (6) in the previous version of the paper was derived from one of these studies [29], but as it does not work well when $\alpha=0$ ($n=1$), as you stated, and adds confusion we have simplified this pressure drop in the new version of the paper saying it scales with ρU_0^2 (see the new equation (6)). This scaling is discussed in a recent publication [30]. Equation (7) is now more clearly derived as discussed in the paper, with the same result that $Ma > k_2$. This threshold clearly divides the cavitating from the non-cavitating impacts in Fig. 3 of the paper for $k_2=0.003$ when the ambient pressure is $P_{amb} = 1$ atm (circular symbols). As mentioned below the value of k_2 decreases when the ambient pressure decreases, which is why cavitation is seen below this critical Ma . The literature shows that air cushioning depends on several properties of the gas, including the gas viscosity, gas speed of sound, gas density (as we show), and even rarefied gas effects. Hence, we agree with the reviewer that equation (7) is too simplistic to fully describe the minimum velocity at which this type of cavitation should occur and that a more full description should take the gas properties into account. As such we have altered the discussion of the pressure on the cylinder surface with air cushioning in paper, which can be found in the paragraph following Eq. (5) through the first full paragraph on the following page. We have also altered the discussion in the Supplemental Material to better address the changes observed in k_2 and k_1 for the different liquids which likely results from the various fluid properties that change when using the different liquids. In summary, when 0 , the air cushioning effect becomes important and cavitation requires a sufficiently high impact velocity ($Ma > k_2$). This threshold depends on both the properties of gas and liquid and has no consensus yet in the current literature. We leave the detailed investigation of this problem for future work.

2.3. The constant k_2 in equation (7) is said to decrease with the ambient pressure. It would be good to comment, based on the values used in this study, how much of an effect the reductions in pressure would have on the constant. It appears that in only the highest ambient pressure studied is cavitation clearly arrested at lower Mach numbers.

Response:

When the ambient pressure equals $P_{amb} = 1$ atm we find that $k_2 \approx 0.003$, but when the ambient pressure is reduced to $P_{amb} \leq 0.5$ atm we find that the constant decreases to $k_2 \approx 0.0022$. This is shown in Fig. 3 of the main text, in which we see that below $Ma \approx 0.0022$ (now indicated by a dotted line) no vaporous cavitation occurs for any of the pressures tested. Additionally, we note

that just to the right of $Ma \approx 0.0022$ few cavitation bubbles are seen in the reduced pressure data with only one or two bubbles appearing in each case (specific numbers not shown in the plot), with the exception of the impact indicated by the one red square for which just over 15 bubbles were seen. We have now drawn a second vertical line (dotted line) to indicate the decrease of k_2 in Fig. 3 and have stated the reduced value of k_2 in the first full paragraph after equation 7 (pg. 4-5).

The reviewer may notice that two of the square data points around $Ma = 0.002$ have been changed from cavitation to no cavitation (orange to gray) in this version of the paper. Because of your question we carefully reviewed the impact videos in this area of the plot and found that what we had previously called cavitation was actually one or two small preexisting bubbles that can be seen to expand after the low pressure wave passes over them. Some of these bubbles remain visible after shrinking and no characteristic shockwave is emitted upon the rebound/collapse. This indicates that these bubbles were not caused by vaporous cavitation, but rather are a simple expansion due to a passing low pressure. We thank the reviewer for this comment, which helped us to catch this mistake.

Editor's comment on 2.

A thorough and convincing discussion of the details of the derivation is prerequisite for publication irrespective of the destination.

Response:

We have now improved the discussion of the derivation of Eq. (7) in the paper which can be found in the paragraph after Eq. (5) and in the paragraph after Eq. (7) (pg. 4-5). As discussed there the pressure on the bottom of a body impacting on a pool with an intervening air layer is a complicated problem that has not been fully resolved in the literature and not all studies agree on how it scales. Hence, we have simplified the derivation and discussed how important parameters such as the fluid properties are likely to affect the pressure on the bottom of an impacting body and the occurrence of cavitation.

3.1. Figure 1 is quite a striking high contrast image. That said, I recommend, in the caption, orienting the reader to the direction of travel of the cylinder prior to impact.

Response:

Thank you for the compliment on this image! We now mention in the figure caption that the cylinder impacts the pool as it falls downward.

3.2. In figure 2d the pressure for the hydrophone appears to drift negatively with time for (a) and (b). Is this drift from the measuring equipment, or was the bath not initially quiescent?

Response:

Thank you for this comment. It alerted us to the fact that we had accidentally used a plot in which the hydrophone data was filtered with a high-pass filter. We have now corrected this so

that the data shown in Fig. 2d is now the raw pressure data from the hydrophone. This removed much of the observed drift in the signal and, as you can now see, both of the signals are quite flat before impact. After the impact several frequencies are generated and a lower frequency causes the signal to decrease below zero, but it rises again with regular oscillations after the time period shown. And yes the pool was initially quiescent.

Editor's comment on 3.

This touches upon the request of referee #2: please provide an improved presentation of data errors, to the benefit of the reader.

Response:

The experimental uncertainties have been discussed, and the omission and mistake caught here have been corrected.

4.3. The sentence "This large tension wave initiates cavitation just beneath the right edge of the cylinder where last contact occurs (Fig. 2a at $t = 34.2 \mu\text{s}$)" I believe is incorrect. Last contact based on figure 2b appears to be earlier than $30 \mu\text{s}$. Based on the position of the wave propagation front in fig. 2b at $22 \mu\text{s}$ it appears to have $\sim 5.6 \text{ mm}$ to reach the right edge, which given the contact line speed of $\sim 1455 \text{ m/s}$ would take $\sim 3.8 \mu\text{s}$. Was the intent to reference (Fig. 2a at $t = 23.8 \mu\text{s}$) here?

Response:

Thank you for catching this mistake. The figure and time reference was actually intended to indicate the time when you can see cavitation under the right edge of the cylinder in Fig. 2a, which is at $t = 34.2 \mu\text{s}$. We see that the wording did not make this very clear. We have now changed the time reference in this sentence to say, " $t = 23.8 - 34.2 \mu\text{s}$ " indicating the time range when both the last contact occurs and the cavitation initiates as stated in the sentence.

4.4. The squared velocity term in equation (3) includes both the contact line and impact velocities. Can the authors comment on this choice?

Response:

Equation (3) comes from the classic work by Wagner [2] who theoretically derived that the pressure on the impacting surface scales with $\rho U_0^2 \alpha^{-1}$, equation (3), and a full description of this pressure term can be found in [2]. We chose this scaling as it is well supported in the literature and applies to our geometry. We noticed that the previous version of the paper did not make it clear that this scaling came from [2], which we now correct in the current version of the paper (see paragraph before (3)).

Editor's comments on 4.

Here the referee points towards possible mistakes.

Response:

The mistake mentioned in comment 4.3 above has been corrected and we have clarified the origin of Eq. (3) as mentioned in comment 4.4.

Reviewer comments, second version:

Reviewer #1 (Remarks to the Author: Overall significance):

The manuscript has been substantially improved. Authors have addressed all my concerns. New series of experiments have been added to the material. And now we can see the edge importance. The manuscript can be published in the present form.

Reviewer #1 (Remarks to the Author: Impact):

Nature Physics
Nature communications

Reviewer #1 (Remarks to the Author: Strength of the claims):

Authors have addressed my concerns.

Reviewer #1 (Remarks to the Author: Reproducibility):

Authors have addressed my concerns.

Reviewer #2 (Remarks to the Author: Overall significance):

Same as my previous comments

Reviewer #2 (Remarks to the Author: Impact):

Same as my previous comments

Reviewer #2 (Remarks to the Author: Strength of the claims):

Same as my previous comments

Reviewer #2 (Remarks to the Author: Reproducibility):

Same as my previous comments

Reviewer #3 (Remarks to the Author: Overall significance):

The current work is now better framed within cavitation literature. The authors provide strong evidence to differentiate the current study from previous low-speed cavitation events. The additional emphasis on the requirement for a sharp edge on the impacting object adds to the novelty of this cavitating system; not only are the thresholds given by the scaling laws required for cavitation, but a geometrical requirement is also present.

Reviewer #3 (Remarks to the Author: Impact):

This paper will spur on new research in impact-induced cavitation. The role that the object geometry plays in altering the cavitation threshold not only highlights the need for future studies but will also influence engineering design based on whether cavitation is desirable or not.

Reviewer #3 (Remarks to the Author: Strength of the claims):

I have no further concerns with this work. The reformulated equation 6 and the 2 corrected square data points in figure 3 shore up the the k_2 derived threshold.

Reviewer #3 (Remarks to the Author: Reproducibility):

Given the level of detail provided in the manuscript and supplementary material, I believe the results of this work can be reproduced by another research group.

From figure 3 it can be determined that 6 different cylinder diameters were used. Given the independence of cavitation and cylinder size, I feel that the presentation of diameter as a range (10-40 mm) while noting proportionality with symbol size is sufficient. I think it would be helpful to reiterate the range of diameters at the end of the second sentence in the caption for figure 3.

Author rebuttal, second version:

**Responses to Reviewer 3's final comment for submission 2
for
Cavitation upon low-speed solid-liquid impact
by**

Nathan Speirs, Kenneth Langley, Zhao Pan, Tadd Truscott, and Sigurdur Thoroddsen

Comment:

From figure 3 it can be determined that 6 different cylinder diameters were used. Given the independence of cavitation and cylinder size, I feel that the presentation of diameter as a range (10-40 mm) while noting proportionality with symbol size is sufficient. I think it would be helpful to reiterate the range of diameters at the end of the second sentence in the caption for figure 3.

Response:

We now state the cylinder diameter range at the end of the second sentence in the caption to figure 3.